# Influence of Seed Soaking and Foliar Application Using Ozonated Water on Two Sweet Pepper Hybrids under Cold Stress

Mohamed A. Sharaf-Eldin [1], Khalid S. Alshallash [2,*], Khadiga R. Alharbi [3,*], Mesfer M. Alqahtani [4], Abdelwahab A. Etman [1], Ali M. Yassin [1], Enas S. Azab [5] and Samira A. F. El-Okkiah [6]

[1] Department of Horticulture, Faculty of Agriculture, Kafrelsheikh University, Kafr Elsheikh 33516, Egypt
[2] College of Science and Humanities-Huraymila, Imam Mohammed Bin Saud Islamic University (IMSIU), Riyadh 11432, Saudi Arabia
[3] Department of Biology, College of Science, Princess Nourah bint Abdulrahman University, Riyadh 11671, Saudi Arabia
[4] Department of Biological Sciences, Faculty of Science and Humanities, Shaqra University, Ad-Dawadimi 11911, Saudi Arabia
[5] Agricultural Botany Department, Faculty of Agriculture, Suez Canal University, Ismailia 41522, Egypt
[6] Department of Agriculture Botany, Faculty of Agriculture, Kafrelsheikh University, Kafr Elsheikh 33516, Egypt
* Correspondence: ksalshallash@imamu.edu.sa (K.S.A.); kralharbi@pnu.edu.sa (K.R.A.)

**Abstract:** The harmful impacts of ozone ($O_3$) on plant development and productivity have been excessively studied. Furthermore, the positive influences of its low concentrations still need to be explored further. The present study was performed to assess the impact of low concentrations of $O_3$ on two sweet pepper hybrids under cold stress. The ozonated water was utilized for seed soaking or foliar application at concentrations of 0, 10, 20, 30, and 40 ppm. Seed soaking using ozonated water for 1 h was compared to soaking in distilled water as a control. Moreover, exogenously ozonated water was sprayed thrice at three-day intervals compared with untreated control. The differences between the applied methods (seed soaking and foliar application using ozonated water) were not statistically detected in most of the evaluated parameters. On the other hand, the evaluated hybrids displayed significant differences in the studied parameters, with the superiority of the Lirica evident in most germination and seedling growth parameters. Both applied methods significantly improved germination and seedling growth parameters. In particular, the concentration of 40 ppm displayed the highest enhancement of the germination index, coefficient velocity, and seedling quality. In addition, it promoted the seedling maintenance of high relative water content (RWC), chlorophyll, proline, and ascorbate peroxidase activity under cold stress conditions. Moreover, it protected the cell wall from damage by decreasing membrane permeability (MP). Generally, the best results were obtained from 40 ppm followed by 30 ppm of $O_3$ as seed soaking or foliar spray. The results pointed out the possible use of $O_3$ in a low concentration to protect the plants from cold stress during germination and early plant growth.

**Keywords:** pepper; priming; ozone; cold stress; physio-biochemical characteristics; morphological traits

## 1. Introduction

Sweet pepper (*Capsicum annuum* L.) is an important vegetable crop extensively cultivated worldwide and with great commercial value [1]. It is utilized as a vegetable, ornamental, and medicinal plant. It has become tremendously popular for its contents of beta-carotene, vitamin C, bioactive compounds, phytochemicals, and polyphenols [2]. It exhibits high genetic diversity in terms of shapes, sizes, antioxidant properties, and biochemical compositions [3]. Furthermore, its genotypes respond differently to the negative impacts of environmental stresses [4–6].

Agricultural production suffers from the detrimental impacts of global climate change, particularly in arid environments [7–9]. The current climate changes are projected to become more frequent and severe [10–12]. Sweet pepper has a tropical origin and requires relatively high temperatures during its growth and development [13]. Low temperature is a widespread environmental factor restricting the production and geographical distribution of most vegetable crops. Cold stress (0–12 °C) has devastating impacts on various biochemical and physiological processes of sweet pepper [14]. It causes membrane leakiness as a result of an inability to raise membrane fluidity and inhibition of photosynthetic processes, thus strongly affecting pepper productivity [15]. These impacts reflect adversely on the production and quality of the plant [16]. Meanwhile, freezing stress (below 0 °C) causes cellular dehydration because of the formation of ice crystals in the extracellular space [17].

Seed germination and seedling establishment are sensitive stages and exhibit a vital role in plant development, growth, and productivity. Hence, it is essential to find effective, affordable, practical, and eco-friendly methods to ameliorate seed germination and seedling growth development. Seed priming is a pre-sowing treatment that is conducted to improve germination and seedling growth [18]. It can stimulate plant performance through rapid and uniform germination, and normal and vigorous seedlings, which has been reported in different crops [19,20]. Additionally, it permits seedlings for development and growth in a wide range of agro-climatic conditions and decreases sensitivity to harmful external factors [21]. Likewise, the exogenous application using biostimulants is a sustainable approach to reinforcing the growth and production of different field crops under environmental stresses [22–24]. Several earlier reports deduced that the exogenous application of different biostimulants ameliorated physiological, biochemical, morphological, yield, and quality parameters of field crops [25–27].

Ozone is a robust oxidizing agent which can virtually react with any biomacromolecule, including proteins, nucleic and fatty acids, and carbohydrates, while it is neither a radical species nor a reactive oxygen species (ROS) [28]. Ozone enters the plant through the stomata, diffuses into the apoplast, and readily decomposes to produce ROS such as superoxide, hydrogen peroxide, hydroxyl radical, and others [29]. Plants primarily deal with $O_3$ stress via an endogenous defensive mechanism consisting of different antioxidant compounds that are able to scavenge free radicals [30]. Thus, $O_3$ could be a beneficial tool for enhancing tolerance to various abiotic stresses without any environmental problems [31]. A detailed literature survey showed that various researchers have evaluated the harmful effect of high $O_3$ concentrations on plants. In this context, González-Fernández et al. [32] elucidated that $O_3$ induced leaf visible injury and reduced leaf photosynthesis, leaf N content, and leaf greenness in spinach and Swiss chard. In addition, $O_3$ impacts on leaves reduced their marketable yield. Moreover, Sharps et al. [33] showed that $O_3$ induced early leaf loss in *Phaseolus vulgaris* and accelerated senescence in *Triticum aestivum*. Similarly, Liang et al. [34] disclosed that $O_3$ reduced the net photosynthetic rate, water use efficiency, kernel weight, number of kernels, and grain yield of maize. Otherwise, studies on the beneficial effects of low ozone concentration in plants are rare.

Ozone is assumed to play an integral role in mitigating the negative impacts of environmental stresses. The impact of seed soaking or exogenous application of low $O_3$ concentration on pepper plants under cold stress is unknown. Accordingly, the objectives of the current study were as follows: to assess different concentrations of $O_3$ in order to identify those that enhanced seed germination and seedling growth under cold stress; to investigate the difference between application methods of seed priming and foliar spraying of seedlings using ozonated water; and to explore the response of two different pepper hybrids to different concentrations or application methods.

## 2. Materials and Methods

### 2.1. Plant Material

Two high-yielding commercial sweet pepper hybrids, namely Zidenka and Lirica, were evaluated under controlled conditions in a growth chamber at the Horticulture

Department, Faculty of Agriculture, Kafrelsheikh University. The studied hybrids were obtained from Rijk Zwaan Company. Healthy pepper seeds were selected with uniform size and shape, sterilized with 1% sodium hypochlorite for 5 min, and then rinsed with distilled water. Later, the seeds from each hybrid were divided into two groups; the first one was utilized to assess the effect of seed soaking in ozone-enriched water on seed germination and seedling parameters. The second one was employed to study the impact of foliar spray with ozonated water on seedling growth parameters when the seedlings had three full-grown leaves. An ozone generator (Matra–GL–2186, Dubendorf, Switzerland) was used to create ozone-enriched water. Based on a preliminary trial that included ozone concentrations from 0 to 100 ppm, the concentrations of 0, 10, 20, 30, and 40 ppm were selected for the present study.

### 2.2. Seed Soaking in Ozonated Water

To study the seed behavior during germination, the seeds from each hybrid were soaked in distilled water for 24 h at room temperature ($25 \pm 2$ °C). After that, they were deeply soaked in distilled water (control) or $O_3$ at the abovementioned concentrations for 1 h. At the end of these treatments, the solutions were removed, and the seeds were washed with distilled water and dried at room temperature for 3 h. Twenty-five seeds from each treatment were placed in 9 cm Petri dishes on two sheets of Whatman No.1 filter paper and moistened with 5 mL distilled water. The dishes were transferred to the incubators at 29 °C, in the dark.

### 2.3. Foliar Application Using Ozonated Water

To study the impact of foliar spray using $O_3$, the un-soaked seeds were sown at seedling trays. The trays were filled with a mixture of peat moss and vermiculite (1:1 *v/v*). At the three full-grown leaves stage of seedling, water enriched with $O_3$ was sprayed on the leaves. The seedlings were sprayed with $O_3$ solutions thrice at three-day intervals. All seedlings were placed in a growth chamber; the temperature was adjusted to 20 °C for 48 h, then reduced to 18 °C during the next 48 h, and then reduced to 12 °C for 72 h. After that, the seedlings were shifted to room temperature. The lighting period was adjusted to 12 h. The samples of transplants were taken at random and separated into shoots and roots to determine the following parameters.

### 2.4. Germination and Seedling Parameters

Germination Percentage (GP) was measured using the following formula which was described by Hafez et al. [35]:

$$GP\ (\%) = \frac{\text{Germinated seeds}}{\text{Total number of seeds}} \times 100$$

Mean Germination Time (MGT) was calculated according to the formula:

$$MGT\ (\text{day}) = \frac{\Sigma nd}{N}$$

where n is the number of germinated seeds on each day, d is the number of days from the beginning of the test, and N is the total number of germinated seeds [35].

Germination Index (GI) was calculated using the following formula [36]:

$$GI\ (\text{seed/day}) = \frac{\text{No. of germinated seeds}}{\text{Day of the first count}} + \cdots + \frac{\text{No.of germinated seeds}}{\text{Day of the final count}}$$

Coefficient velocity (CV) was calculated according to Edwards and Sundstrom [36]:

$$CV\ (\%) = \frac{1}{MGT} \times 100$$

where MGT is the mean time germination.

The roots and shoots from 10 seedlings (cotyledon stage 10-day-old seedlings) of each treatment were cut and their fresh weights were measured. Then, the obtained roots and shoots were dried at 70 °C for 48 h, and their dry weights were recorded. The shoot diameter was measured using a vernier at 1 cm above the medium surface.

### 2.5. Physiological Parameters

Chlorophyll content in the first fully expanded leaves without destroying them was determined by a SPADE-501 chlorophyll meter (Konica Minolta, Osaka, Japan). The chlorophyll meter was employed to determine leaf greenness [37] Membrane permeability (MP) was determined by estimating electrolyte leakage according to Valentovic et al. [38]. Five plants randomly chosen per replicate were cut into uniformly sized discs and 0.5 g of the discs were taken from the middle portion of the fully developed youngest leaf and washed with distilled water to remove surface contamination. The discs were placed in closed tubes containing 20 mL of deionized water and incubated at 25 °C for 24 h. The electrical conductivity (EC, mg/L) of the bathing solution (EC1) was determined. The samples were then boiled in the water bath at 120 °C for 20 min and the electrical conductivity (EC2) was determined after cooling the solution to room temperature. Membrane permeability was calculated as a ratio of (EC1/EC2) $\times$ 100.

Relative water content (RWC) was recorded following Anjum et al. [39]. Fully developed leaves were cut and weighed directly to record the fresh weight (FW) and then soaked in the deionized distilled water for 16 h. The excess surface water was dried with paper towels to determine the turgid weight (TW) of the leaf. The leaves were then oven-dried to constant weight for the dry weight (DW). The RWC was determined according to the formula RWC $= \frac{FW-DW}{TW-DW} \times 100$.

Proline content ($\mu$mol g$^{-1}$) was estimated following Bates et al. [40]. An amount of 0.5 g fresh leaves was uninformed into 10 mL of 3% aqueous sulfosalicylic acid. Two ml of the extract was added to two ml of acid ninhydrin and two ml of glacial acetic acid. The mixture was put in a boiling water bath for an hour. The reaction ended in an ice bath. To extract the organic part, 4 mL of toluene was added to the mixture and a UV-visible spectrophotometer was utilized at 520 nm for measurements, and toluene was used as a blank. The activity of ascorbate peroxidase (APX, $\mu$mol g$^{-1}$min$^{-1}$) was assayed as described by Nakano and Asada [41]. An amount of 0.5 g from fresh leaves was uninformed at 0–4 °C in 3 mL of 50 mM TRIS at pH 7.8, containing 1 mM EDTA-Na2 and 7.5% polyvinylpyrrolidone. The solution was centrifuged at 1200 rpm for 20 min at 4 °C. A spectrophotometer (UV-160A, Shimadzu, Japan) was used to measure total soluble enzyme activities. Three ml from the reaction mixture included 50 Mm phosphate buffer (pH 7.8), 0.1 mM EDTA, 0.5 mM ascorbate, 0.1 mM H$_2$O$_2$, and 0.1 mL enzyme extract. H$_2$O$_2$ was added to start the reaction, and then ascorbate oxidation was measured at 290 nm for 3 min. The molar extinction coefficient for ascorbate (2.8 mM$^{-1}$ cm$^{-1}$) was used to quantify the enzyme activity.

### 2.6. Data Analysis

The experimental layout was factorial in a completely randomized design (CRD) in three replicates. Statistical procedures were applied employing R statistical software version 4.1.1 (Foundation for Statistical Computing, Vienna, Austria). The mean values of treatments were compared according to the Tukey HSD test at a 0.01 level. The experiments were repeated twice, and the presented results were the mean of the two experiments.

### 3. Results

#### 3.1. Germination Parameters

The evaluated hybrids did not significantly differ in germination percentage (GP) (Table 1). Otherwise, the hybrids displayed significant differences in mean germination time (MGT), germination index (GI), and coefficient velocity (CV). Zidenka had the highest

MGT while Lirica had the more elevated GI and CV (Table 1). Seed soaking in ozonated water did not affect the GP, while it significantly impacted the MGT, GI, and CV. Increasing $O_3$ concentration in the water solution from 0 to 40 ppm decreased the MGT, and the highest interaction effect was observed in Zidenka soaked in 10 ppm $O_3$. On the other hand, increasing $O_3$ concentration increased the GI and CV with the 40 ppm application showing the highest values. The highest interaction effect of GI and CV was recorded for Lirica soaked in 40 ppm $O_3$.

**Table 1.** Impact of seed soaking in ozonated water at different concentrations on germination parameters of two pepper hybrids.

| Studied Factor | | GP (%) | MGT (Day) | GI (Seed/Day) | CV (%) |
|---|---|---|---|---|---|
| Pepper hybrid (H) | | | | | |
| | Zidenka | 92.26 | 7.31 [a] | 6.35 [b] | 13.72 [b] |
| | Lirica | 92.00 | 6.79 [b] | 7.22 [a] | 14.91 [a] |
| Seed soaking (S) | | | | | |
| | 0 ppm $O_3$ | 92.00 | 7.29 [b] | 5.73 [d] | 13.71 [d] |
| | 10 ppm $O_3$ | 92.00 | 8.06 [a] | 6.76 [e] | 12.41 [e] |
| | 20 ppm $O_3$ | 91.33 | 7.02 [c] | 6.71 [c] | 14.26 [c] |
| | 30 ppm $O_3$ | 92.66 | 6.55 [d] | 8.04 [b] | 15.32 [b] |
| | 40 ppm $O_3$ | 92.66 | 6.34 [e] | 8.68 [a] | 15.86 [a] |
| Interaction | | | | | |
| | 0 ppm $O_3$ | 92.00 | 7.29 [b] | 5.73 [c] | 13.72 [e] |
| | 10 ppm $O_3$ | 92.00 | 8.06 [a] | 4.77 [d] | 12.41 [f] |
| Zidenka | 20 ppm $O_3$ | 90.67 | 7.02 [c] | 6.17 [c] | 13.57 [e] |
| | 30 ppm $O_3$ | 92.00 | 6.56 [d] | 7.22 [b] | 14.23 [d] |
| | 40 ppm $O_3$ | 93.33 | 6.34 [e] | 7.87 [b] | 14.68 [c] |
| | 0 ppm $O_3$ | 92.00 | 7.29 [b] | 5.73 [c] | 13.72 [e] |
| | 10 ppm $O_3$ | 92.00 | 8.06 [a] | 4.77 [d] | 12.41 [f] |
| Lirica | 20 ppm $O_3$ | 92.00 | 6.68 [d] | 7.25 [b] | 14.97 [c] |
| | 30 ppm $O_3$ | 93.33 | 6.09 [e] | 8.87 [a] | 16.43 [b] |
| | 40 ppm $O_3$ | 93.33 | 5.87 [f] | 9.48 [a] | 17.04 [a] |
| ANOVA | df | | | | |
| H | 1 | 0.539 [NS] | <0.001 | <0.001 | <0.001 |
| S | 4 | 0.284 [NS] | <0.001 | <0.001 | <0.001 |
| H × S | 4 | 0.237 [NS] | <0.001 | <0.001 | <0.001 |

Seed soaking in ozonated water at concentrations of 0, 10, 20, 30, and 40 ppm for 1 h. NS: Not significant, Means followed by different letters under the same factor are significantly different according to Tukey's HSD test ($p \leq 0.01$). GP: germination percentage, MGT: mean germination time (day), GI: germination index, and CV: coefficient velocity.

### 3.2. Fresh and Dry Weight

The heavier roots and shoots fresh weight were produced by the Lirica hybrid compared to Zidenka (Table 2). Otherwise, the highest root/shoot ratio was observed for Zidenka. Seed soaking in ozonated water at 30 ppm recorded the highest value of root fresh weight followed by the application of 40 ppm. The highest interaction impact was observed for Lirica soaked in 40 ppm followed by Zidenka soaked in 10 ppm. In the case of shoot fresh weight, 40 ppm of $O_3$ achieved the highest value followed by 30 ppm. The highest fresh weight of the shoot was exhibited by Lirica soaked in 40 and 30 ppm respectively. The total weight of pepper seedlings was recorded for the application of 40 ppm followed by 30 ppm, particularly using the hybrid Lirica.

Concerning dry weight, there were non-significant differences between the two studied hybrids in the root and shoot dry weights as well as root/shoot ratio. In addition, $O_3$ concentrations did not significantly differ in the case of roots and total seedling dry weight. On the other hand, the shoots produced from seeds treated with 40 ppm significantly exceeded all treatments in dry weight, especially the hybrid Zidenka.

**Table 2.** Impact of seed soaking in ozonated water at different concentrations on the seedling fresh and dry weight of two pepper hybrids.

| Studied Factor | | Fresh Weight (g/Plant) | | | Dry Weight (g/Plant) | | |
|---|---|---|---|---|---|---|---|
| | | Roots | Shoots | Root/Shoot Ratio | Roots | Shoots | Root/Shoot Ratio |
| Pepper hybrids | | | | | | | |
| Zidenka | | 0.141 [b] | 0.211 [b] | 0.668 [a] | 0.009 | 0.016 | 0.563 |
| Lirica | | 0.150 [a] | 0.228 [a] | 0.658 [b] | 0.014 | 0.015 | 0.933 |
| Seed soaking | | | | | | | |
| 0 ppm $O_3$ | | 0.141 [c] | 0.205 [e] | 0.688 [b] | 0.021 | 0.015 [b] | 1.400 |
| 10 ppm $O_3$ | | 0.147 [b] | 0.195 [d] | 0.754 [a] | 0.009 | 0.016 [b] | 0.563 |
| 20 ppm $O_3$ | | 0.136 [d] | 0.219 [c] | 0.621 [d] | 0.007 | 0.014 [c] | 0.500 |
| 30 ppm $O_3$ | | 0.147 [a] | 0.232 [b] | 0.634 [c] | 0.008 | 0.015 [b] | 0.53 |
| 40 ppm $O_3$ | | 0.155 [b] | 0.245 [a] | 0.633 [c] | 0.012 | 0.017 [a] | 0.706 |
| Interaction | | | | | | | |
| | 0 ppm $O_3$ | 0.133 [g] | 0.192 [i] | 0.693 [b] | 0.009 | 0.013 [c] | 0.692 |
| | 10 ppm $O_3$ | 0.159 [b] | 0.193 [i] | 0.824 [a] | 0.010 | 0.018 [a] | 0.556 |
| Zidenka | 20 ppm $O_3$ | 0.132 [g] | 0.211 [g] | 0.626 [e] | 0.009 | 0.015 [b] | 0.600 |
| | 30 ppm $O_3$ | 0.140 [e] | 0.224 [e] | 0.625 [e] | 0.009 | 0.016 [b] | 0.563 |
| | 40 ppm $O_3$ | 0.142 [e] | 0.236 [c] | 0.602 [f] | 0.012 | 0.018 [a] | 0.667 |
| | 0 ppm $O_3$ | 0.151 [d] | 0.219 [f] | 0.689 [b] | 0.033 | 0.018 [a] | 1.833 |
| | 10 ppm $O_3$ | 0.135 [f] | 0.198 [h] | 0.682 [b] | 0.009 | 0.013 [c] | 0.692 |
| Lirica | 20 ppm $O_3$ | 0.141 [e] | 0.227 [d] | 0.621 [e] | 0.007 | 0.013 [c] | 0.538 |
| | 30 ppm $O_3$ | 0.155 [c] | 0.241 [b] | 0.643 [d] | 0.008 | 0.014 [c] | 0.57 |
| | 40 ppm $O_3$ | 0.169 [a] | 0.255 [a] | 0.663 [c] | 0.008 | 0.014 [c] | 0.571 |
| ANOVA | df | | | *p* value | | | |
| H | 1 | <0.001 | <0.001 | <0.001 | 0.387 [NS] | <0.001 | 0.375 [NS] |
| S | 4 | <0.001 | <0.001 | <0.001 | 0.423 [NS] | <0.001 | 0.388 [NS] |
| H × S | 4 | <0.001 | <0.001 | <0.001 | 0.411 [NS] | <0.001 | 0.557 [NS] |

Seed soaking in ozonated water at concentrations of 0, 10, 20, 30, and 40 ppm for 1 h. NS: Not significant, Means followed by different letters under the same factor are significantly different according to Tukey's HSD test ($p \leq 0.01$).

### 3.3. Seedling Measurements

The hybrid Lirica produced a higher stem diameter and stem length compared to Zdenka under both soaking treatments or foliar application (Table 3). Among the evaluated treatments, 40 ppm $O_3$ concentration using both application methods showed the maximum stem diameter and stem length. On the other hand, in all cases the minimum value was produced by the untreated control. The results indicated that there is no significant difference between the general means of the application methods (seed soaking and foliar spray).

The hybrid Zidenka exhibited a higher root length than Lirica (Table 3). In all cases, the untreated plants had the shortest roots. The seeds soaked at a rate of 40 ppm exhibited the highest root length, while the seedlings sprayed with 30 ppm recorded the highest values. In general, the tallest seedling was obtained from 40 ppm $O_3$ treatment either with seed soaking or foliar spray (Table 3). The seedling spray exhibited statistically higher root and stem length compared to seed soaking.

The hybrid Zidenka exhibited the highest root fresh weight under seed soaking treatment while Lirica had the highest root fresh weight using the foliar application (Table 4). Meanwhile, Lirica under soaking application displayed the highest shoot fresh weight and total fresh weight of seedlings under the foliar application. There were significant differences among the concentration of either seed soaking or foliar spray treatments for root, shoot, and total fresh weight of pepper seedlings. In all cases, the 40 ppm $O_3$ treatment was the best for the aforementioned parameters. The two treatments, soaking and foliar application, did not exhibit significant differences except for root fresh weight reflecting the superiority of the soaking treatment (Table 4).

**Table 3.** Impact of seed soaking or foliar spray using ozonated water on seedlings at different concentrations: stem diameter and seedling length of two pepper hybrids grown under cold stress.

| Studied Factor | | Stem Diameter (mm) | | Root Length (cm) | | Stem Length (cm) | |
|---|---|---|---|---|---|---|---|
| | | Soaking | Spray | Soaking | Spray | Soaking | Spray |
| Pepper hybrid (H) | | | | | | | |
| Zidenka | | 2.66 [b] | 2.65 [b] | 8.17 [a] | 8.25 [a] | 11.70 [b] | 12.11 |
| Lirica | | 2.71 [a] | 2.71 [a] | 7.90 [b] | 7.80 [b] | 12.21 [a] | 12.21 |
| Seed treatment (T) | | | | | | | |
| 0 ppm $O_3$ | | 2.55 [e] | 2.55 [e] | 7.26 [e] | 7.26 [d] | 10.90 [e] | 12.18 [bc] |
| 10 ppm $O_3$ | | 2.77 [b] | 2.69 [b] | 8.26 [b] | 7.62 [c] | 12.30 [b] | 12.30 [b] |
| 20 ppm $O_3$ | | 2.61 [d] | 2.60 [d] | 7.62 [d] | 8.00 [b] | 11.80 [d] | 12.06 [cd] |
| 30 ppm $O_3$ | | 2.65 [c] | 2.65 [c] | 8.00 [c] | 9.03 [a] | 11.96 [c] | 11.93 [d] |
| 40 ppm $O_3$ | | 2.83 [a] | 2.87 [a] | 9.03 [a] | 7.98 [b] | 12.81 [a] | 12.60 [a] |
| Mean (T) | | 2.67 | 2.66 | 7.68 [B] | 8.02 [A] | 11.88 [B] | 12.09 [A] |
| Interaction | | | | | | | |
| | 0 ppm $O_3$ | 2.520 [f] | 2.520 [f] | 7.000 [f] | 7.00 [i] | 12.000 [cd] | 12.000 [cd] |
| | 10 ppm $O_3$ | 2.900 [b] | 2.733 [c] | 8.167 [b] | 8.40 [c] | 9.900 [f] | 12.467 [b] |
| Zidenka | 20 ppm $O_3$ | 2.593 [g] | 2.587 [e] | 7.233 [e] | 7.75 [g] | 11.600 [e] | 12.133 [c] |
| | 30 ppm $O_3$ | 2.617 [ef] | 2.613 [de] | 7.633 [d] | 8.20 [d] | 12.133 [cd] | 12.063 [cd] |
| | 40 ppm $O_3$ | 2.700 [c] | 2.780 [b] | 8.133 [b] | 9.50 [a] | 12.867 [a] | 12.600 [ab] |
| | 0 ppm $O_3$ | 2.597 [f] | 2.597 [e] | 7.533 [d] | 7.53 [h] | 12.600 [b] | 12.433 [b] |
| | 10 ppm $O_3$ | 2.650 [d] | 2.650 [d] | 8.467 [a] | 8.13 [e] | 11.900 [cd] | 11.900 [cd] |
| Lirica | 20 ppm $O_3$ | 2.630 [de] | 2.630 [de] | 7.500 [d] | 7.50 [h] | 12.000 [cd] | 12.000 [cd] |
| | 30 ppm $O_3$ | 2.700 [c] | 2.700 [c] | 7.833 [c] | 7.80 [f] | 11.800 [e] | 11.800 [d] |
| | 40 ppm $O_3$ | 2.973 [a] | 2.973 [a] | 8.167 [b] | 8.57 [b] | 12.767 [ab] | 12.767 [a] |
| ANOVA | df | | | *p* value | | | |
| H | 1 | <0.001 | <0.001 | <0.001 | <0.001 | <0.001 | <0.001 |
| S | 4 | <0.001 | <0.001 | <0.001 | <0.001 | <0.001 | <0.001 |
| H × S | 4 | <0.001 | <0.001 | <0.001 | <0.001 | <0.001 | <0.001 |

Seed soaking in ozonated water at concentrations of 0, 10, 20, 30, and 40 ppm for 1 h. Foliar application of seedlings using ozonated water at the three full-grown leaves stage thrice at three-day intervals. Means followed by different letters under the same factor are significantly different according to Tukey's HSD test ($p \leq 0.01$).

**Table 4.** Impact of seed soaking or foliar spray using ozonated water at different concentrations on seedling fresh weight of two pepper hybrids grown under cold stress.

| Studied Factor | | Root Fresh Weight (g) | | Shoot Fresh Weight (g) | | Seedling Fresh Weight (g) | |
|---|---|---|---|---|---|---|---|
| | | Soaking | Spray | Soaking | Spray | Soaking | Spray |
| Pepper hybrid (H) | | | | | | | |
| Zidenka | | 0.797 [a] | 0.740 [b] | 1.76 [b] | 1.794 | 2.56 | 2.528 [b] |
| Lirica | | 0.774 [b] | 0.775 [a] | 1.78 [a] | 1.784 | 2.55 | 2.558 [a] |
| Seed treatment (T) | | | | | | | |
| 0 ppm $O_3$ | | 0.743 [c] | 0.743 [c] | 1.77 [b] | 1.773 [b] | 2.51 [c] | 2.51 [b] |
| 10 ppm $O_3$ | | 0.756 [c] | 0.680 [d] | 1.71 [d] | 1.733 [b] | 2.46 [e] | 2.45 [c] |
| 20 ppm $O_3$ | | 0.752 [c] | 0.733 [c] | 1.73 [c] | 1.728 [d] | 2.48 [d] | 2.46 [c] |
| 30 ppm $O_3$ | | 0.801 [b] | 0.776 [b] | 1.74 [c] | 1.743 [c] | 2.54 [b] | 2.52 [b] |
| 40 ppm $O_3$ | | 0.876 [a] | 0.850 [a] | 1.91 [a] | 1.916 [a] | 2.79 [a] | 2.76 [a] |
| Mean (T) | | 0.770 [A] | 0.752 [B] | 1.76 | 1.77 | 2.53 | 2.52 |
| Interaction | | | | | | | |
| | 0 ppm $O_3$ | 0.710 [f] | 0.710 [e] | 1.760 [d] | 1.760 [d] | 2.470 [ef] | 2.470 [f] |
| | 10 ppm $O_3$ | 0.820 [c] | 0.667 [f] | 1.660 [f] | 1.793 [c] | 2.480 [e] | 2.460 [f] |
| Zidenka | 20 ppm $O_3$ | 0.750 [e] | 0.713 [e] | 1.767 [d] | 1.757 [d] | 2.517 [d] | 2.470 [f] |
| | 30 ppm $O_3$ | 0.803 [c] | 0.753 [d] | 1.783 [c] | 1.780 [c] | 2.587 [c] | 2.533 [d] |
| | 40 ppm $O_3$ | 0.903 [a] | 0.850 [a] | 1.863 [b] | 1.860 [b] | 2.767 [b] | 2.710 [b] |
| | 0 ppm $O_3$ | 0.777 [d] | 0.777 [a] | 1.787 [c] | 1.787 [c] | 2.563 [c] | 2.563 [c] |
| | 10 ppm $O_3$ | 0.693 [f] | 0.693 [c] | 1.753 [d] | 1.753 [d] | 2.447 [ef] | 2.447 [f] |
| Lirica | 20 ppm $O_3$ | 0.753 [e] | 0.753 [e] | 1.700 [e] | 1.700 [e] | 2.453 [f] | 2.453 [f] |
| | 30 ppm $O_3$ | 0.800 [c] | 0.800 [b] | 1.707 [e] | 1.707 [e] | 2.507 [d] | 2.507 [e] |
| | 40 ppm $O_3$ | 0.850 [b] | 0.850 [a] | 1.973 [a] | 1.973 [a] | 2.823 [a] | 2.823 [a] |
| ANOVA | df | | | *p* value | | | |
| H | 1 | <0.001 | <0.001 | <0.001 | <0.001 | <0.001 | <0.001 |
| S | 4 | <0.001 | <0.001 | <0.001 | <0.001 | <0.001 | <0.001 |
| H × S | 4 | <0.001 | <0.001 | <0.001 | <0.001 | <0.001 | <0.001 |

Seed soaking in ozonated water at concentrations of 0, 10, 20, 30, and 40 ppm for 1 h. Foliar application of seedlings using ozonated water at the three full-grown leaves stage thrice at three-day intervals. Means followed by different letters under the same factor are significantly different according to Tukey's HSD test ($p \leq 0.01$).

Lirica produced higher root and total seedling dry weight than Zedinka under both treatments, soaking and foliar spray (Table 5). Otherwise, Zidenka had the highest shoot dry weight under both treatments. The $O_3$ concentrations displayed significant differences in both applications (seed soaking and seedling spray) for dry matter partitioning of pepper seedlings. The concentration of 40 ppm $O_3$ was more effective in increasing dry matter content compared to the other treatments. However, the untreated control had the lowest values (Table 5).

**Table 5.** Impact of seed soaking or foliar spray using ozonated water on seedling dry weight of two pepper hybrids under cold stress.

| Studied Factor | | Root Dry Weight (g) | | Shoot Dry Weight (g) | | Seedling Dry Weight (g) | |
|---|---|---|---|---|---|---|---|
| | | Soaking | Spray | Soaking | Spray | Soaking | Spray |
| Pepper hybrid (H) | | | | | | | |
| Zidenka | | 0.055 b | 0.055 b | 0.182 a | 0.183 a | 0.237 b | 0.237 b |
| Lirica | | 0.061 a | 0.061 a | 0.179 b | 0.179 b | 0.241 a | 0.240 a |
| Seed treatment (T) | | | | | | | |
| 0 ppm $O_3$ | | 0.053 d | 0.053 d | 0.175 d | 0.174 d | 0.226 d | 0.223 e |
| 10 ppm $O_3$ | | 0.055 c | 0.053 d | 0.181 b | 0.188 b | 0.237 b | 0.241 b |
| 20 ppm $O_3$ | | 0.058 b | 0.055 c | 0.170 e | 0.167 e | 0.229 c | 0.229 d |
| 30 ppm $O_3$ | | 0.067 a | 0.058 b | 0.177 c | 0.175 c | 0.236 b | 0.233 c |
| 40 ppm $O_3$ | | 0.052 e | 0.068 a | 0.200 a | 0.201 a | 0.267 a | 0.269 a |
| Mean (T) | | 0.057 | 0.057 | 0.180 | 0.181 | 0.239 | 0.239 |
| Interaction | | | | | | | |
| | 0 ppm $O_3$ | 0.052 e | 0.052 g | 0.174 f | 0.174 e | 0.226 f | 0.226 f |
| | 10 ppm $O_3$ | 0.060 c | 0.057 e | 0.182 c | 0.195 b | 0.242 c | 0.252 c |
| Zidenka | 20 ppm $O_3$ | 0.052 e | 0.052 g | 0.176 e | 0.169 f | 0.228 f | 0.221 g |
| | 30 ppm $O_3$ | 0.053 e | 0.052 g | 0.181 c | 0.175 e | 0.234 e | 0.227 f |
| | 40 ppm $O_3$ | 0.060 c | 0.061 c | 0.199 b | 0.202 a | 0.259 b | 0.263 b |
| | 0 ppm $O_3$ | 0.055 d | 0.055 f | 0.178 d | 0.178 d | 0.233 e | 0.233 e |
| | 10 ppm $O_3$ | 0.051 e | 0.051 g | 0.181 c | 0.181 c | 0.231 e | 0.231 e |
| Lirica | 20 ppm $O_3$ | 0.059 c | 0.059 d | 0.165 g | 0.165 e | 0.225 f | 0.225 f |
| | 30 ppm $O_3$ | 0.065 b | 0.065 b | 0.174 f | 0.174 g | 0.239 d | 0.239 d |
| | 40 ppm $O_3$ | 0.075 a | 0.075 a | 0.201 a | 0.201 a | 0.276 a | 0.276 a |
| ANOVA | df | | | *p* value | | | |
| H | 1 | <0.001 | <0.001 | <0.001 | <0.001 | <0.001 | <0.001 |
| S | 4 | <0.001 | <0.001 | <0.001 | <0.001 | <0.001 | <0.001 |
| H × S | 4 | <0.001 | <0.001 | <0.001 | <0.001 | <0.001 | <0.001 |

Seed soaking in ozonated water at concentrations of 0, 10, 20, 30, and 40 ppm for 1 h. Foliar application of seedlings using ozonated water at the three full-grown leaves stage thrice at three-day intervals. Means followed by different letters under the same factor are significantly different according to Tukey's HSD test ($p \leq 0.01$).

### 3.4. Physiological Parameters

The chlorophyll content and RWC of Lirica leaves under both treatments (seed soaking and foliar spray) were significantly higher than those of Zidenka (Table 6). The evaluated hybrids did not significantly differ in MP under both application methods. The $O_3$ concentration in both treatments displayed significant differences in chlorophyll content, MP, and RWC. The concentration of 40 ppm $O_3$ in the soaking method followed by 30 ppm displayed the highest chlorophyll content and RWC and the lowest MP. Likewise, the concentration of 30 ppm in the foliar application followed by 40 ppm exhibited the highest chlorophyll content and RWC and the lowest MP. The untreated control displayed the lowest chlorophyll content and RWC and the highest MP. In general, the seed soaking method was more effective than foliar spray which produced higher chlorophyll content and RWC (Table 6).

**Table 6.** Impact of seed soaking or foliar spray using ozonated water at different concentrations on some physiological parameters of two pepper hybrids grown under cold stress.

| Studied Factor | | Chlorophyll (SPAD Value) | | Membrane Permeability (MP %) | | Relative Water Content (RWC %) | |
|---|---|---|---|---|---|---|---|
| | | Soaking | Spray | Soaking | Spray | Soaking | Spray |
| Pepper hybrid (H) | | | | | | | |
| Zidenka | | 39.67 [b] | 38.48 [b] | 16.32 | 16.07 | 75.74 [b] | 76.25 [b] |
| Lirica | | 43.76 [a] | 40.73 [a] | 16.30 | 16.07 | 77.78 [a] | 78.21 [a] |
| Seed treatment (T) | | | | | | | |
| 0 ppm $O_3$ | | 43.20 [d] | 37.71 [c] | 18.08 [a] | 18.08 [a] | 72.75 [e] | 72.95 [e] |
| 10 ppm $O_3$ | | 37.85 [e] | 37.68 [c] | 16.46 [c] | 17.31 [b] | 72.65 [c] | 74.06 [d] |
| 20 ppm $O_3$ | | 43.43 [c] | 40.13 [b] | 17.31 [b] | 15.43 [c] | 74.40 [d] | 76.96 [c] |
| 30 ppm $O_3$ | | 44.15 [b] | 41.93 [a] | 15.43 [d] | 14.26 [e] | 77.43 [b] | 81.95 [a] |
| 40 ppm $O_3$ | | 46.95 [a] | 40.11 [b] | 14.26 [e] | 14.88 [d] | 82.60 [a] | 81.16 [b] |
| Mean (T) | | 41.72 [A] | 39.51 [B] | 16.31 [A] | 16.07 [B] | 77.01 [B] | 77.23 [A] |
| Interaction | | | | | | | |
| | 0 ppm $O_3$ | 34.20 [i] | 34.30 [g] | 18.233 [a] | 18.233 [a] | 71.20 [h] | 71.50 [i] |
| | 10 ppm $O_3$ | 34.80 [h] | 39.60 [d] | 16.367 [c] | 16.367 [c] | 76.60 [d] | 76.13 [d] |
| Zidenka | 20 ppm $O_3$ | 41.13 [f] | 36.47 [f] | 17.233 [b] | 17.233 [b] | 73.33 [g] | 73.07 [h] |
| | 30 ppm $O_3$ | 42.10 [d] | 40.13 [cd] | 15.200 [e] | 15.200 [e] | 76.23 [d] | 75.70 [e] |
| | 40 ppm $O_3$ | 46.13 [c] | 41.77 [a] | 14.467 [f] | 14.467 [f] | 81.37 [b] | 81.00 [b] |
| | 0 ppm $O_3$ | 41.50 [e] | 41.13 [b] | 17.933 [a] | 17.933 [a] | 74.30 [f] | 74.40 [g] |
| | 10 ppm $O_3$ | 37.60 [g] | 40.60 [c] | 16.567 [c] | 16.567 [c] | 76.70 [d] | 76.50 [d] |
| Lirica | 20 ppm $O_3$ | 45.73 [c] | 38.90 [e] | 17.400 [b] | 17.400 [b] | 75.47 [e] | 75.07 [f] |
| | 30 ppm $O_3$ | 46.20 [b] | 40.13 [cd] | 15.667 [d] | 15.667 [d] | 78.63 [c] | 78.23 [c] |
| | 40 ppm $O_3$ | 47.77 [a] | 42.10 [a] | 14.067 [g] | 14.067 [g] | 83.83 [a] | 82.90 [a] |
| ANOVA | df | | | *p* value | | | |
| H | 1 | <0.001 | <0.001 | 0.521 [NS] | 0.495 [NS] | <0.001 | <0.001 |
| S | 4 | <0.001 | <0.001 | <0.001 | <0.001 | <0.001 | <0.001 |
| H × S | 4 | <0.001 | <0.001 | <0.001 | <0.001 | <0.001 | <0.001 |

Seed soaking in ozonated water at concentrations of 0, 10, 20, 30, and 40 ppm for 1 h. Foliar application of seedling using ozonated water at the three full-grown leaves stage thrice at three-day intervals. NS: Not significant, Means followed by different letters under the same factor are significantly different according to Tukey's HSD test ($p \leq 0.01$).

Lirica recorded the highest proline content and ascorbate peroxides under both treatments, seed soaking and foliar spray (Table 7). Proline content and ascorbate peroxides were increased significantly with increasing $O_3$ concentration in both treatments. The highest values were assigned for 40 ppm followed by 30 ppm, while the untreated control gave the lowest values (Table 7). The application methods, seed soaking or foliar spray, did not display a significant difference in proline content and ascorbate peroxides.

*3.5. Interrelationship among the Evaluated Treatments and Parameters*

Exploring the interrelationship between the assessed treatments and parameters is a vital aspect that can provide valuable information. The heatmap is an appropriate statistical procedure to study the association among investigated treatments and parameters. In the present study, the heatmap and hierarchical clustering based on the studied seedling traits and physiological parameters divided the assessed treatments into different clusters (Figure 1). In both hybrids, the concentration of 40 ppm followed by 30 ppm using either one of the application methods possessed the highest values for most evaluated traits (depicted in blue). On the contrary, the untreated control of Lirica had the lowest values (depicted in red).

**Table 7.** Proline content and ascorbate peroxides in pepper plants as affected by evaluated hybrids and $O_3$ treatments, seed soaking or foliar spray, at different concentrations.

| Studied Factor | | Proline Content ($\mu$mol g$^{-1}$) | | Ascorbate Peroxides ($\mu$mol g$^{-1}$ min$^{-1}$) | |
|---|---|---|---|---|---|
| | | Soaking | Spray | Soaking | Spray |
| Pepper hybrid (H) | | | | | |
| Zidenka | | 0.959 [b] | 0.961 [b] | 0.399 [b] | 0.399 [b] |
| Lirica | | 0.964 [a] | 0.968 [a] | 0.400 [a] | 0.402 [a] |
| Seed treatment (T) | | | | | |
| 0 ppm $O_3$ | | 0.917 [d] | 0.935 [d] | 0.381 [d] | 0.389 [c] |
| 10 ppm $O_3$ | | 0.927 [c] | 0.918 [c] | 0.386 [c] | 0.382 [d] |
| 20 ppm $O_3$ | | 0.864 [e] | 0.865 [e] | 0.359 [e] | 0.359 [e] |
| 30 ppm $O_3$ | | 0.967 [b] | 0.967 [b] | 0.402 [b] | 0.402 [b] |
| 40 ppm $O_3$ | | 1.132 [a] | 1.140 [a] | 0.471 [a] | 0.472 [a] |
| Mean (T) | | 0.961 | 0.965 | 0.400 | 0.401 |
| Interaction | | | | | |
| | 0 ppm $O_3$ | 0.912 [g] | 0.913 [f] | 0.379 [g] | 0.379 [f] |
| | 10 ppm $O_3$ | 0.922 [f] | 0.923 [e] | 0.383 [f] | 0.384 [e] |
| Zidenka | 20 ppm $O_3$ | 0.912 [g] | 0.910 [f] | 0.379 [g] | 0.378 [f] |
| | 30 ppm $O_3$ | 0.981 [c] | 0.979 [c] | 0.408 [c] | 0.407 [c] |
| | 40 ppm $O_3$ | 1.070 [b] | 1.075 [b] | 0.445 [b] | 0.447 [b] |
| | 0 ppm $O_3$ | 0.922 [f] | 0.957 [d] | 0.383 [f] | 0.398 [d] |
| | 10 ppm $O_3$ | 0.933 [e] | 0.913 [f] | 0.388 [e] | 0.379 [f] |
| Lirica | 20 ppm $O_3$ | 0.817 [h] | 0.819 [g] | 0.340 [h] | 0.341 [g] |
| | 30 ppm $O_3$ | 0.953 [d] | 0.955 [d] | 0.396 [d] | 0.397 [d] |
| | 40 ppm $O_3$ | 1.194 [a] | 1.197 [a] | 0.496 [a] | 0.497 [a] |
| ANOVA | df | | | | |
| H | 1 | <0.001 | <0.001 | <0.001 | <0.001 |
| S | 4 | <0.001 | <0.001 | <0.001 | <0.001 |
| H × S | 4 | <0.001 | <0.001 | <0.001 | <0.001 |

Seed soaking in ozonated water at concentrations of 0, 10, 20, 30, and 40 ppm for 1 h. Foliar application of seedlings using ozonated water at the three full-grown leaves stage thrice at three-day intervals. Means followed by different letters under the same factor are significantly different according to Tukey's HSD test ($p \leq 0.01$).

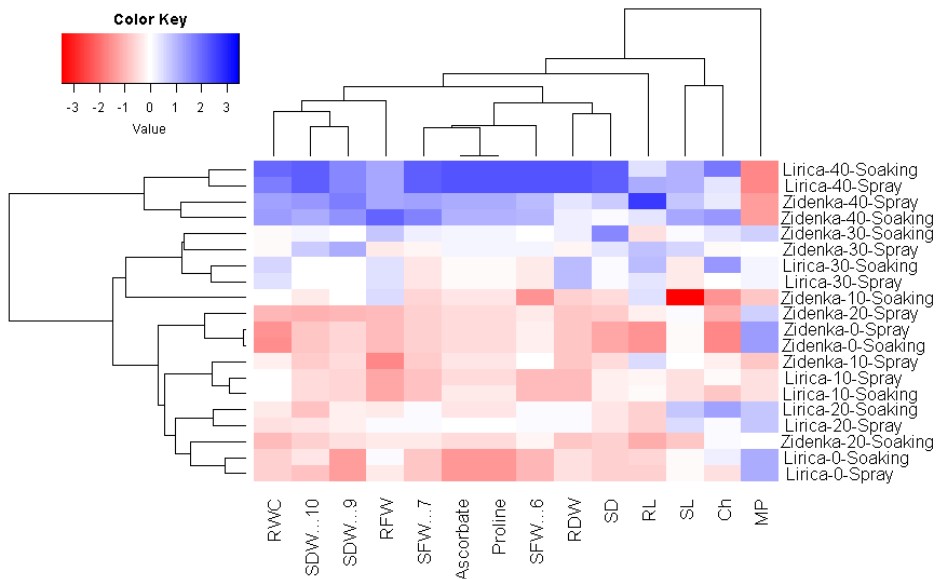

**Figure 1.** Heatmap and hierarchical clustering divide the assessed pepper hybrids, seed soaking, and foliar application using ozonated water into different clusters based on seedling and physiological parameters. Blue and red colors indicate high and low values for the corresponding trait, respectively. SD: stem diameter, RL: root length, SL: stem length, RFW: root fresh weight, SFW: shoot fresh weight, SFW: seedling fresh weight, RDW: root dry weight, SDW: shoot dry weight, SDW: seedling dry weight, Ch: chlorophyll content, MP: membrane permeability, RWC: relative water content, Proline: proline content, and Ascorbate: ascorbate peroxides.

## 4. Discussion

The production of vegetable crops under stressful conditions becomes a great challenge, particularly under conditions of climate change. Pepper plants are adversely impacted by different biotic and abiotic stress conditions [42,43]. Sweet pepper is of tropical origin and is a warm-season crop. Therefore, cold stress has a severely adverse impact on plant development and production [44]. The present study was conducted to explore the impact of low concentrations of $O_3$ on two sweet pepper hybrids under cold stress. The evaluated hybrids displayed considerable differences in most studied germination and seedling parameters. Generally, Lirica exhibited stronger seedling growth, indicating that it had a better adaptation to cold stress than Zidenka. Lirica's stronger seedlings appeared with higher biomass production, dry matter, and stem length and diameter. The darker greenness of its leaf expressed as SPAD could be the reason for this enhancement via increasing photosynthesis processes. It also could retain more relative water and proline content, and its APX activity was higher under cold stress, which allows it to cope with the harmful effect of cold stress. Likewise, Lozo et al. [5], ElShamey et al. [23], Penella et al. [45] detected significant differences among sweet papers grown under abiotic stresses.

Seed priming and foliar anti-stressors are important methods for alleviating stress on the plant and were reported using different substances [46–51]. The negative effect of $O_3$ on plant development and production was extensively studied [52–54]. However, it has several beneficial properties when it is applied in low concentrations. It degenerates quickly into oxygen, hence it is considered environmentally safe and has a non-toxic residue [55]. Due to its strong oxidative ability in both aqueous and gaseous phases, it is used as an effective anti-microbial agent. Moreover, it is useful for fruit preservation because it has no side effects on nutritional values and product quality [56]. It has gained much attention due to its crucial impact in enhancing secondary metabolites and bioactive properties which help ameliorate plant growth and productivity and work against many pathogens [57]. The positive impacts of exogenous $O_3$ in mitigating stressful conditions of salinity [58], drought [59], and cold stress [60] have been reported. Yet, studies on the role of low $O_3$ concentrations on physiological and biochemical adjustments during seed germination and early growth periods of sweet pepper plants are still lacking.

The obtained results confirmed that soaking pepper seeds in enriched water with $O_3$, particularly at 40 ppm, seemed to improve germination parameters i.e., GI, CV, MGT, and fresh and dry weights. These improvements may be due to the role of $O_3$ in ROS induction inside the seeds [61,62]. Consequently, when ROS accumulates in a suitable amount, it can be helpful for seed germination and seedling growth by adjusting cellular growth and enhancing antioxidant activity. In this respect, Bhattacharjee [63] and Desoky et al. [64] demonstrated that ROS are important signaling molecules in biotic stress responses, where they serve as messengers for the activation of defense genes. Antioxidants by themselves act as scavengers for excess ROS production in seed biology, which plays a vital role in the growth processes occurring at early embryogenesis during seed development. Moreover, $O_3$ participates in the mechanisms underlying radicle protrusion during seed germination and seed aging [65,66]. It can activate the acclimation process and provide an improvement in plant tolerance to subsequent stress. These results are supported by those obtained by [67] and [68]. On the other side, extreme cold stress or high $O_3$ concentration, especially with a too-long exposure time, can damage the cells which are considered unfavorable for seed germination and seedling growth. The resultant ROS has been considered for too-long exposure time as being very harmful, which often leads to oxidative stress at the cellular level [65]. Moreover, they can disrupt cellular membranes, and lead to enzyme inactivation, protein degradation, and ionic imbalance [69]. Nevertheless, a short time exposure was most beneficial. Similarly, Sudhakar et al. [61], Abeli et al. [70], Avdeeva et al. [71], and Rodrigues et al. [72] elucidated the positive impacts of low concentrations of $O_3$ on the germination and seedling parameters of tomato, alpine, wheat, and soybean, respectively.

The applied methods (seed soaking and foliar application using ozonated water) did not exhibit considerable differences in most evaluated parameters. Both application meth-

ods, especially at 40 ppm, enhanced the growth parameters of pepper seedlings compared with the untreated control. It seems that the low concentrations of $O_3$ (10–40 ppm) have protective effects against the chilling stress on pepper seedlings. These results agreed with the results of [61]. Ozone as a strong oxidative factor has adverse impacts on plant photosynthesis [62]. Otherwise, in the present study, low concentrations for a short time of exposure enhanced chlorophyll content.

In this context, Flowers et al. [29] elucidated that the plants exposed to relatively low $O_3$ concentrations for a short time possessed higher chlorophyll intensity in comparison to untreated controls. On the contrary, several studies pointed out the negative effect of high $O_3$ doses or exposure on plants, such as potatoes [73], beans [29], and cowpea [74]. In this context, Wilkinson et al. [75] suggested that the decrease in growth was associated with longer exposure to higher $O_3$ concentrations, which can be a reason for the reduction in carbon transport to the roots, and nutrient and water uptake.

The results displayed that exposure to low $O_3$ concentrations decreased membrane permeability. The reduction in permeability could protect the cells from damage during environmental stresses, including cold stress. It was considered a sensitive stress marker due to the sensitivity of cellular membranes to different environmental stresses that always cause oxidative damage accompanied by increasing permeability [76]. Moreover, the obtained results showed that the low concentrations of $O_3$ increased the RWC, APX, and proline, which could help to preserve the leaf water and nutrient content against cold stress. Plants prone to chilling stress often show symptoms of water stress through the inhibition of water absorption and water loss. The water loss is primarily related to the loss of membrane properties or the transfer of membranes from the state of the natural fluid to a restricted, less liquid, and semi-crystalline state [77]. RWC is a quantitative indicator of the water status of plants. Plant cells have a group of antioxidants (both enzymatic and non-enzymatic) to reduce the stress caused by ROS [78]. In addition, ROS can oxidize lipids and affect the normal membrane function [79,80]. Proline is a non-enzymatic antioxidant, which accumulates in response to biotic and abiotic stresses [81]. Furthermore, it helps to save osmotic adjustment and membrane stability, and decrease the harmful effects of ROS as a free radical scavenger [82]. High concentrations of proline likely have a role in stress tolerance under low temperatures by preserving the physical characteristics to protect both membranes and proteins [79,80]. Under cold stress, proline accumulation as inert compatible osmolytes can protect subcellular structures and macromolecules [83]. Moreover, it can protect protein integrity and enhance the activities of different enzymes, which works in this case as a molecular chaperone [84].

Under abiotic stress such as chilling, drought, or salinity, antioxidant systems in plants act to prevent or alleviate the membrane peroxidation produced by ROS [85]. Ascorbate peroxidase scavenging ROS in plants has an essential role in the detoxification or scavenging of $H_2O_2$ ascorbate–glutathione cycles [86]. APX activity increased in different plants when exposed to different abiotic stresses [87], e.g., $O_3$ exposure [88]. Similarly, Umponstira et al. [89] and Yan et al. [90] elucidated that APX activity depended on $O_3$ dose and exposure period, while low enzyme activity was obtained with a high $O_3$ dose with long exposure time. Thus, the observed increase in APX activity as an antioxidant enzyme could be attributed to the low $O_3$ concentrations and the short exposure time. It may be noted that the effects of $O_3$ stress on the antioxidant enzymes are complicated and depend on the plant species, treatment intensity, and exposure time.

## 5. Conclusions

The positive influences of $O_3$ on cultivated plants still need more investigation under normal or stressful conditions. The present study confirmed that $O_3$ in a low concentration could be effectively employed to enhance pepper germination and seedling growth against the damaging effects of cold stress. The application of seed soaking or seedling foliar spray using ozonated water enhanced germination and seedling growth parameters by elevating the antioxidant status of pepper plants compared with the untreated controls. The two

applied methods (seed soaking and foliar application using ozonated water) did not display significant differences in most of the evaluated parameters. Meanwhile, the evaluated hybrids exhibited significant differences in the studied parameters, with Lirica showing superiority in most germination and seedling growth parameters. Among the tested concentrations, 30 and 40 ppm of ozonated water as seed soaking or foliar spray displayed the highest favorable impacts on germination and seedling growth. The utilization of ozonated water has a promising future in enhancing field crops against environmental stresses due to its affordability, practicality, and eco-friendly properties.

**Author Contributions:** Conceptualization, M.A.S.-E., A.A.E., A.M.Y., E.S.A. and S.A.F.E.-O.; methodology, M.A.S.-E., A.A.E., A.M.Y., E.S.A. and S.A.F.E.-O.; software, K.S.A., K.R.A. and E.S.A.; validation, M.A.S.-E., K.S.A., K.R.A., A.A.E., A.M.Y. and E.S.A.; formal analysis, M.A.S.-E., A.A.E., A.M.Y., E.S.A. and S.A.F.E.-O.; investigation, M.A.S.-E., A.A.E., A.M.Y., E.S.A. and S.A.F.E.-O.; resources, K.S.A., K.R.A. and M.M.A.; data curation M.A.S.-E., K.S.A., K.R.A., A.A.E., A.M.Y., E.S.A. and S.A.F.E.-O.; writing—original draft preparation, K.S.A., K.R.A., M.M.A., A.A.E., A.M.Y. and E.S.A.; writing—review and editing, M.A.S.-E., K.S.A., K.R.A., M.M.A., A.A.E., A.M.Y. and E.S.A.; visualization, A.A.E., A.M.Y. and E.S.A.; supervision, A.A.E., A.M.Y. and E.S.A.; funding acquisition, K.S.A., K.R.A. and M.M.A. All authors have read and agreed to the published version of the manuscript.

**Funding:** This research was funded by Princess Nourah bint Abdulrahman University Researchers Supporting Project number (PNURSP2022R188), Princess Nourah bint Abdulrahman University, Riyadh, Saudi Arabia.

**Institutional Review Board Statement:** Not applicable.

**Informed Consent Statement:** Not applicable.

**Data Availability Statement:** The data presented in this study are available upon request from the corresponding author.

**Acknowledgments:** The co-author Khalid S. Alshallash from Saudi Arabia would like to thank the deanship of scientific research at Imam Mohammed Bin Saud Islamic University. Also, the co-author Mesfer M. Alqahtani from Saudi Arabia would like to thank the deanship of scientific research at Shaqra University for supporting publishing this research work. In addition, the co-author Khadiga R. Alharbi would like to thank Princess Nourah bint Abdulrahman University for funding this work through the Researchers Supporting Project number (PNURSP2022R188), Princess Nourah bint Abdulrahman University, Riyadh, Saudi Arabia.

**Conflicts of Interest:** The authors declare no conflict of interest.

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
