# Peer review of "Influence of Seed Soaking and Foliar Application Using Ozonated Water on Two Sweet Pepper Hybrids under Cold Stress"

_sustainability, doi:10.3390/su142013453_

Round 1
Reviewer 1 Report
The manuscript entitled "Influence of Seed Soaking and Foliar Application using Ozonated Water on Germination, Seedling Vigor, and Physiological Parameters of Two Sweet Pepper Hybrids under Cold Stress" investigates the effects of low O3 concentration on seed germination and seedling growth under cold stress and what applications will more efficient seed priming vs. foliar spray of the seedling. The data present in the study concluded that ozone in a low concentration could be effectively employed to enhance pepper germination and seedlings' growth against the damaging effects of cold stress. The most suitable dose was 30 and 40 ppm of ozonated water as seed soaking or foliar spray. The study is on a topic of relevance and general interest to the journal's readers. I found the paper to be overall well written. However, I have several concerns about the manuscript that should be addressed before publication.
· The authors must carefully read the manuscript to correct typos and grammar to improve the manuscript.
· Any abbreviation must be associated with the full name at the first mention in the abstract and main text, then just use the abbreviation.
· In the material and methods section, all the chemicals, software, and equipment sources must be added or completed (add city, state, and country).
· Add a reference to lines 74-76
· Line 145 (C1/C2) should be EC1/EC2
· The authors need to add a brief description of the calculation methods of proline and ascorbate peroxidase
· Line 180-181 is repeated for the previous sentence and should be removed
Author Response
Dear Editor,
We would like to thank you and the reviewers for the time and efforts devoted to our manuscript entitled “Influence of Seed Soaking and Foliar Application using Ozonated Water on Germination, Seedling Vigor, and Physiological Parameters of Two Sweet Pepper Hybrids under Cold Stress” (Sustainability-1931613). We have revised the manuscript according to the comments and suggestions pointed out by the reviewers. We have addressed the comments of the reviewers point-by-point below in red color; in addition, we have highlighted all the associated changes made to the manuscript using track changes.
Yours sincerely,
Authors
Reviewer 1
The manuscript entitled "Influence of Seed Soaking and Foliar Application using Ozonated Water on Germination, Seedling Vigor, and Physiological Parameters of Two Sweet Pepper Hybrids under Cold Stress" investigates the effects of low O3 concentration on seed germination and seedling growth under cold stress and what applications will more efficient seed priming vs. foliar spray of the seedling. The data present in the study concluded that ozone in a low concentration could be effectively employed to enhance pepper germination and seedlings' growth against the damaging effects of cold stress. The most suitable dose was 30 and 40 ppm of ozonated water as seed soaking or foliar spray. The study is on a topic of relevance and general interest to the journal's readers. I found the paper to be overall well written. However, I have several concerns about the manuscript that should be addressed before publication.
Re: We would like to thank Reviewer 1 for his time dedicated to our manuscript. We greatly appreciate his positive assessment of our work and constructive comments for improving our manuscript.
- The authors must carefully read the manuscript to correct typos and grammar to improve the manuscript.
Re: The manuscript has been carefully revised and the language has been considerably improved.
- Any abbreviation must be associated with the full name at the first mention in the abstract and main text, then just use the abbreviation.
Re: All abbreviations were revised throughout the manuscript and presented at the first mention as full name and abbreviation and then just the abbreviations were used.
- In the material and methods section, all the chemicals, software, and equipment sources must be added or completed (add city, state, and country).
Re: Done as suggested (lines 146, 191, 219, 228)
- Add a reference to lines 74-76
Re: References have been added as suggested (lines 115-120)
- Line 145 (C1/C2) should be EC1/EC2
Re: Thanks so much for your accurate revision, modified as suggested ( line 202)
- The authors need to add a brief description of the calculation methods of proline and ascorbate peroxidase
Re: More details have been added as suggested (lines 209-224)
- Line 180-181 is repeated for the previous sentence and should be removed
Re: The sentence corresponding with seed soaking treatment while the previous one related to the evaluated hybrids.
Thanks so much for your revision, we sincerely appreciate all the valuable comments and suggestions which helped us to improve the quality of our manuscript.
Reviewer 2 Report
Dear editor
The manuscript should be improved based on the below comments:
The Title in too long. please revise it
The abstract must be including all summary parts of manuscript.
The necessity of subject is not in introduction clearly. Please re-write it.
The literature review is key section in introduction. The manuscript should be re-write based on main studies in the world
Why do you do this method? fitness, challenges and etc?
The discussion is the main achievement the study, so should be discuss about:
the main fitness than other similar studies?
What is the main achievement to plant physiology? in plant conservation? Environmental management?
The manuscript showing some shortcomings. After some revisions should be revised.
Best Regards
Author Response
Dear Editor,
We would like to thank you and the reviewers for the time and efforts devoted to our manuscript entitled “Influence of Seed Soaking and Foliar Application using Ozonated Water on Germination, Seedling Vigor, and Physiological Parameters of Two Sweet Pepper Hybrids under Cold Stress” (Sustainability-1931613). We have revised the manuscript according to the comments and suggestions pointed out by the reviewers. We have addressed the comments of the reviewers point-by-point below in red color; in addition, we have highlighted all the associated changes made to the manuscript using track changes.
Yours sincerely,
Authors
Reviewer 2
Dear Editor
The manuscript should be improved based on the below comments:
Re: We would like to thank Reviewer 2 for his time dedicated to our manuscript. We greatly appreciate the comment provided by the reviewer to improve the quality of the manuscript.
The title is too long. please revise it
Re: The title has been modified to be shorter as suggested
The abstract must be including all summary parts of the manuscript.
Re: The abstract has been revised and improved as suggested
The necessity of subject is not in introduction clearly. Please re-write it.
Re: The importance of the studied subject has been clarified as requested (lines 84-125)
The literature review is key section in introduction. The manuscript should be re-write based on main studies in the world. Why do you do this method? fitness, challenges and etc?
Re: The literature has been added and more details have been clarified in the introduction (lines 115-127)
The discussion is the main achievement the study, so should be discussed about:
the main fitness than other similar studies?
What is the main achievement to plant physiology? in plant conservation? Environmental management?
Re: The discussion section has been re-written and considerably improved as suggested
The manuscript showing some shortcomings. After some revisions should be revised.
Best Regards
Re: All issues made by the reviewer have been carefully addressed, and the manuscript has been considerably improved. Thanks so much for your thoughtful comments and efforts toward improving our manuscript.
Reviewer 3 Report
Comments to the authors
General comments:
#1 The purpose of this research was to shed light on a few questions. For optimal seed germination and plant growth while exposed to cold, what ozone concentration should be used? The question is whether seed priming or foliar spraying with ozonated water on young seedlings is the superior application strategy. Does the stage of development matter when deciding how much ozone to use? Is there a difference in how pepper hybrids react to various doses and applications?
#2 The work as a whole has a good scientific quality, however, the author has to focus on polishing the language because some of the phrases lack coherence and logic. Overall, the work is of good quality. In addition to that, I noticed several instances of duplication throughout the write-up. To improve the overall quality of the document, I suggest sending it to an editor who specializes in the English language.
#3 It is recommended not to start the sentence with an abbreviation.
#4 Because the information about pepper plants, their economic worth, and their relevance was not included, the introduction section could incorporate further information about these topics.
#5 At the very end of the section under "Introduction," objectives need to be outlined very explicitly. In the scenario that the authors wish to present the objectives in the form of questions, the conclusion is where they should make their replies abundantly clear.
#6 Materials and methods should contain and detail all of the experiments. For instance, the authors should describe why these two sweet pepper hybrids were selected. If they checked before for their cold stress or other stress factors. How they grow pepper plants in the seedling trays, including irrigation and other agricultural practices, and so on.
#7 In statistical analysis, please explain how the factors are distributed!
#8 In the section under "Materials and Methods," I think it'd be beneficial to divide the treatments into two groups: ozone treatments and cold treatments. This would make the treatments more understandable. It is also recommended to divide the Materials and methods to subsections for example, under physiological parameters, chlorophyll analysis, EC, RWC, etc…
#9 The results section has a lot of tables; in all, there are 7 tables and one figure with not much explanation. I would suggest to the authors that they read through the figures and only include the ones that are significant to the conclusion of the study in their work. Providing the readers with a large number of tables will not assist them to achieve the goals. The roots/shoots ratio can be added to the examined parameters as a good indicator of stress tolerance. In the tables or under the table as *, information regarding the treatments, such as soaking time and foliar spray timings, may be added. I recommend the authors replace the interrelationship among the treatments and parameters with correlation analysis. It will give more valuable points compared with the heatmap
#10 Discussion section is well stated; nonetheless, there should be additional pepper-related sources added. Significant results showed be discussed more.
#11 Conclusion should answer the objectives questions clearly. I recommend the authors rewrite the conclusion.
Other comments:
Line 23: kindly, change “Otherwise” to “Furthermore”
Line 36: the only spot mentioning biochemical characteristics, might be morphological and physiological.
Line 36: kindly, change “characters” to “characteristic”.
Line 61: I suggest adding some literature about the foliar spray.
Line 73-77: The authors mentioned “A detailed literature survey….” The citation should be added.
Line 93: “at transplanting age” which age?
Line 100: “dried at room temperature” for how long!
Lines 112: kindly replace “next” with the following and delete the sentence starting with “ the next parameters…..”
Line 114: Kindly be constantly writing the subtitles whether all in capital or small letters.
Line 115: “ Germination Percentage (GP) was measured using the following formula which described by ….[18] and kindly remove 18 from line 116.
Line 117: Please change Mean germination time (MGT) to Mean Germination Time (MGT)
Line 125: the mean time to germination, kindly delete “to”.
Line 127: kindly change 10 days old to 10-day-old seedlings.
Lines 131-132: repeated information
Line 177: how the authors analyzed the GP, the significance/letters are not displayed in table 1.
Line 210 (table 2). As root biomass is one of the determining factors for stress tolerance, the root/shoots ratio may be substituted for seedlings.
Author Response
Dear Editor,
We would like to thank you and the reviewers for the time and efforts devoted to our manuscript entitled “Influence of Seed Soaking and Foliar Application using Ozonated Water on Germination, Seedling Vigor, and Physiological Parameters of Two Sweet Pepper Hybrids under Cold Stress” (Sustainability-1931613). We have revised the manuscript according to the comments and suggestions pointed out by the reviewers. We have addressed the comments of the reviewers point-by-point below in red color; in addition, we have highlighted all the associated changes made to the manuscript using track changes.
Yours sincerely,
Authors
Reviewer 3
General comments:
#1 The purpose of this research was to shed light on a few questions. For optimal seed germination and plant growth while exposed to cold, what ozone concentration should be used? The question is whether seed priming or foliar spraying with ozonated water on young seedlings is the superior application strategy. Does the stage of development matter when deciding how much ozone to use? Is there a difference in how pepper hybrids react to various doses and applications?
#2 The work as a whole has a good scientific quality, however, the author has to focus on polishing the language because some of the phrases lack coherence and logic. Overall, the work is of good quality. In addition to that, I noticed several instances of duplication throughout the write-up. To improve the overall quality of the document, I suggest sending it to an editor who specializes in the English language.
Re: We would like to thank Reviewer 3 for his time dedicated to our manuscript. We highly appreciate his comment to improve the quality of the manuscript. The manuscript has been revised by our colleague working at Florida University, USA.
#3 It is recommended not to start the sentence with an abbreviation.
Re: The manuscript has been revised and starting the sentence with an abbreviation has been avoided.
#4 Because the information about pepper plants, their economic worth, and their relevance was not included, the introduction section could incorporate further information about these topics.
Re: The importance of pepper has been extended (lines 51-58)
#5 At the very end of the section under "Introduction," objectives need to be outlined very explicitly. In the scenario that the authors wish to present the objectives in the form of questions, the conclusion is where they should make their replies abundantly clear.
Re: The objectives have been revised (lines 127-134)
#6 Materials and methods should contain and detail all of the experiments. For instance, the authors should describe why these two sweet pepper hybrids were selected. If they checked before for their cold stress or other stress factors. How they grow pepper plants in the seedling trays, including irrigation and other agricultural practices, and so on.
Re: The used pepper hybrids are high-yielding commercial hybrids, therefore they have been evaluated. The materials and method section has been revised and more details have been added.
#7 In statistical analysis, please explain how the factors are distributed!
Re: This is a controlled trial, therefore, he applied experimental design was factorial in Completely Randomized Design (CRD) in three replicates (line 228).
#8 In the section under "Materials and Methods," I think it'd be beneficial to divide the treatments into two groups: ozone treatments and cold treatments. This would make the treatments more understandable. It is also recommended to divide the Materials and methods to subsections for example, under physiological parameters, chlorophyll analysis, EC, RWC, etc…
Re: The materials and methods section has been divided into subsections as requested.
#9 The results section has a lot of tables; in all, there are 7 tables and one figure with not much explanation. I would suggest to the authors that they read through the figures and only include the ones that are significant to the conclusion of the study in their work. Providing the readers with a large number of tables will not assist them to achieve the goals. The roots/shoots ratio can be added to the examined parameters as a good indicator of stress tolerance. In the tables or under the table as *, information regarding the treatments, such as soaking time and foliar spray timings, may be added. I recommend the authors replace the interrelationship among the treatments and parameters with correlation analysis. It will give more valuable points compared with the heatmap
Re: Requested details have been added as footnotes for all tables. Using letters to demonstrate statistical significance facilitates following the obtained results in the tables. After your permission, we would like to keep the heatmap as it explores the interrelationship among the assessed treatments and simultaneously among the studied parameters. It clearly divided the assessed treatments into different clusters, the concentration of 40 ppm followed by 30 ppm in both application methods displayed the highest values for most evaluated traits (depicted in blue) while untreated control had the lowest values (depicted in red).
#10 Discussion section is well stated; nonetheless, there should be additional pepper-related sources added. Significant results showed be discussed more.
Re: The discussion section has been revised and more related results have been added
#11 Conclusion should answer the objectives questions clearly. I recommend the authors rewrite the conclusion.
Re: The conclusion has been modified to answer the addressed objectives.
Other comments:
Line 23: kindly, change “Otherwise” to “Furthermore”
Re: Done as requested (line 25)
Line 36: the only spot mentioning biochemical characteristics, might be morphological and physiological.
Re: Modified as suggested (lines 47-48)
Line 36: kindly, change “characters” to “characteristic”.
Re: Modified as suggested (lines 47)
Line 61: I suggest adding some literature about foliar spray.
Re: More details have been added to the foliar application (lines 97-101)
Line 73-77: The authors mentioned “A detailed literature survey….” The citation should be added.
Re: The citations have been added (lines 115-122)
Line 100: “dried at room temperature” for how long!
Re: The duration has been added (line 157)
Lines 112: kindly replace “next” with the following and delete the sentence starting with “ the next parameters…..”
Re: Done as suggested (line 170-171)
Line 114: Kindly be constantly writing the subtitles whether all in capital or small letters.
Re: All subtitles have been revised and modified as suggested
Line 115: “ Germination Percentage (GP) was measured using the following formula which described by ….[18] and kindly remove 18 from line 116.
Re: Modified as requested (line 173)
Line 117: Please change Mean germination time (MGT) to Mean Germination Time (MGT)
Re: Modified as requested (line 175)
Line 125: the mean time to germination, kindly delete “to”.
Re: Modified as requested (line 183)
Line 127: kindly change 10 days old to 10-day-old seedlings.
Re: Modified as requested (line 184)
Lines 131-132: repeated information
Re: The repeated sentence has been deleted (lines 188-190), thanks so much for your accurate revision.
Line 177: how the authors analyzed the GP, the significance/letters are not displayed in table 1.
Re: In Table 1, the values of GP are presented without letters revealing no statistically significant difference between hybrids and also among the evaluated application methods as indicated in the ANOVA analysis at the end of table 1.
Line 210 (table 2). As root biomass is one of the determining factors for stress tolerance, the root/shoots ratio may be substituted for seedlings.
Re: The root/shoot ratio has been added as suggested (Table 2)
Thanks so much for your thoughtful comments and efforts toward improving our manuscript
Reviewer 4 Report
The manuscript „Influence of Seed Soaking and Foliar Application using Ozonated Water on Germination, Seedling Vigor, and Physiological Parameters of Two Sweet Pepper Hybrids under Cold Stress” is an original paper explaining the impact of low concentrations of ozone (10 - 40 ppm) on sweet pepper hybrids. The authors pointed out the possible use of ozone in a low concentration to protect the plants from cold stress during germination and early plant growth.
The article is well organized, and the submission is worth publication. However, there are a few flaws that must be solved. I would like to recommend to the authors to major revise the manuscript according to the comments described below.
Introduction
Line 39-42 - please change your minds, avoiding repetition of words „requires”, „germination”
Line 46-48 - as above, please change the sentences avoiding the repetition of words „ Low temperature”
Line 66-69 - Please correct the sense and construction of the sentence.
Line 73-75 - please add the appropriate literature
In introduction, I also recommend adding a few sentences about the features / properties of the pepper varieties used
Methods
Please include the final unit in all formulas.
Line 134-136 - please detail the description of the determination of chlorophyll or add the relevant literature.
Line 145 - Please explain what the letter C means in the formula C1/C2
On what basis ozone concentrations 10-40 ppm were chosen, please explain.
Results:
Table 7 . In what units are the values in Table 7 (proline and ascorbate peroxides content)
Discusion
In the opinion of the reviewer, in discussion too little attention was paid to the obtained results. There is absolutely no mention of the difference between the two used varieties - from which could be the reason for these parametric differences - please complete this
It is not described how the individual growth parameters of pepper cultivars after ozone treatment differed from other ozone-treated plants. – please, add it.
Conslusion
Line 410-411 - A statement taken too far. Only 4 concentrations were tested in the experiment, so it cannot be said that the optimal dose of ozone is 30 and 40 ppm. Please correct / lighten this statement
Author Response
Dear Editor,
We would like to thank you and the reviewers for the time and efforts devoted to our manuscript entitled “Influence of Seed Soaking and Foliar Application using Ozonated Water on Germination, Seedling Vigor, and Physiological Parameters of Two Sweet Pepper Hybrids under Cold Stress” (Sustainability-1931613). We have revised the manuscript according to the comments and suggestions pointed out by the reviewers. We have addressed the comments of the reviewers point-by-point below in red color; in addition, we have highlighted all the associated changes made to the manuscript using track changes.
Yours sincerely,
Authors
Reviewer 4
The manuscript “Influence of Seed Soaking and Foliar Application using Ozonated Water on Germination, Seedling Vigor, and Physiological Parameters of Two Sweet Pepper Hybrids under Cold Stress” is an original paper explaining the impact of low concentrations of ozone (10 - 40 ppm) on sweet pepper hybrids. The authors pointed out the possible use of ozone in a low concentration to protect the plants from cold stress during germination and early plant growth.
The article is well organized, and the submission is worth publication. However, there are a few flaws that must be solved. I would like to recommend to the authors to major revise the manuscript according to the comments described below.
Re: We would like to thank Reviewer 4 for his time dedicated to our manuscript. We greatly appreciate his positive assessment of our work, encouraging words, and constructive comments for improving our manuscript.
Introduction
Line 39-42 - Please change your minds, avoiding the repetition of words “requires”, “germination”
Re: The paragraph has been revised and modified as requested (lines 51-65)
Line 46-48 - as above, please change the sentences avoiding the repetition of words “Low temperature”
Re: All the paragraph has been revised and modified as requested (64-82)
Line 66-69 - Please correct the sense and construction of the sentence.
Re: The sentence has been rephrased (lines line 107-112)
Line 73-75 - please add the appropriate literature
Re: Appropriate literature has been added (lines 115-122)
In introduction, I also recommend adding a few sentences about the features/properties of the pepper varieties used
Re: Characterization of pepper genotypes has been added to the introduction (lines 56-58)
Methods
Please include the final unit in all formulas.
Re: The units have been added to all formulas as requested
Line 134-136 - please detail the description of the determination of chlorophyll or add the relevant literature.
Re: More details and references have been added (lines 192-195)
Line 145 - Please explain what the letter C means in the formula C1/C2
Re: The formula has been revised and corrected (200-204)
On what basis ozone concentrations 10-40 ppm were chosen, please explain.
Re: More explanation has been added as requested (149-151).
Results
Table 7. In what units are the values in Table 7 (proline and ascorbate peroxides content)
Re: The units have been added as requested (all tables)
Discussion
In the opinion of the reviewer, in the discussion too little attention was paid to the obtained results. There is absolutely no mention of the difference between the two used varieties - from which could be the reason for these parametric differences - please complete this
Re: More explanations have been added for the varietal variations (lines 396-407) and all discussion section has been revised and improved.
Conclusion
Line 410-411 - A statement taken too far. Only 4 concentrations were tested in the experiment, so it cannot be said that the optimal dose of ozone is 30 and 40 ppm. Please correct / lighten this statement
Re: The sentence has been modified as suggested (508-510) and all the conclusion has been revised and improved.
Round 2
Reviewer 3 Report
Comments to the authors
Thank the authors for responding to the comments.
Minor comments
Line 143: Kindly change the “p” of the percentage in a capital letter and remove the “:”.
Line 149: Kindly change the “i” of the index to a capital letter.
Line 170: Kindly it is not preferable to start the sentence with the abbreviation “MP was calculated as a ratio..”
Line 151: Kindly change the “v” of velocity in a capital letter.
Line 171: Please change to Relative Water Content (RWC).
Table 1: Kindly add “ns” on the GP and explain in the footer.
Table 2: Kindly add “ns” on the root/shoot ratio of the dry weight and explain it in the footer.
Table 2: Please adjust the format to 0 ppm O3.
Table 6: Please change (MP %) and (RWC %).
Line 326: Please change “O3 treatments”.
Author Response
Dear Editor,
We would like to thank you and the reviewers for the time and efforts devoted to our manuscript entitled “Influence of Seed Soaking and Foliar Application using Ozonated Water on Germination, Seedling Vigor, and Physiological Parameters of Two Sweet Pepper Hybrids under Cold Stress” (Sustainability-1931613). We have revised the manuscript according to the comments and suggestions pointed out by the reviewers. We have addressed the comments of the reviewers point-by-point below in red color; in addition, we have highlighted all the associated changes made to the manuscript using track changes.
Yours sincerely,
Authors
Reviewer 3
Thank the authors for responding to the comments.
Re: We would like to thank Reviewer 1 for his time dedicated to our manuscript.
Minor comments
Line 143: Kindly change the “p” of the percentage in a capital letter and remove the “:”.
Re: Done as requested (please see line 146)
Line 149: Kindly change the “i” of the index to a capital letter.
Re: Done as suggested (line 152)
Line 170: Kindly it is not preferable to start the sentence with the abbreviation “MP was calculated as a ratio..”
Re: “MP” has been changed to “Membrane permeability” (line 174)
Line 151: Kindly change the “v” of velocity in a capital letter.
Re: Done as requested (line 154)
Line 171: Please change to Relative Water Content (RWC).
Re: Done as requested (line 176)
Table 1: Kindly add “ns” on the GP and explain in the footer.
Re: Done as requested in Tables 1,2, 6
Table 2: Kindly add “ns” on the root/shoot ratio of the dry weight and explain it in the footer.
Re: Done as requested
Table 2: Please adjust the format to 0 ppm O3.
Re: Done as requested
Table 6: Please change (MP %) and (RWC %).
Re: Done as requested
Line 326: Please change “O3 treatments”.
Re: Done as requested (line 333)
Thanks so much for your time and efforts dedicated to our manuscript
Reviewer 4 Report
The reviewer appreciates the effort made for the revision of the manuscript. Authors significantly improved the manuscript. The paper can be accepted without any further changes.
Author Response
Reviewer 4
The reviewer appreciates the effort made for the revision of the manuscript. The authors significantly improved the manuscript. The paper can be accepted without any further changes.
Re: We would like to thank Reviewer 4 for his time dedicated to our manuscript.